# Distribution of Irrigated and Rainfed Agricultural Land in a Semi-Arid Sandy Area

**Huihui Zheng** [1], **Zhiting Sang** [1], **Kaige Wang** [1], **Yan Xu** [1,2,3,*] **and Zhaoyang Cai** [1]

1   College of Land Science and Technology, China Agricultural University, Beijing 100193, China
2   Key Laboratory of Agricultural Land Quality and Monitoring of Nature Resource, Beijing 100193, China
3   Land Use and Management Center, China Agricultural University, Beijing 100193, China
*    Correspondence: xyan@cau.edu.cn

**Abstract:** Under water resource and terrain constraints, a certain scale threshold of irrigated and rainfed agricultural areas exists in semi-arid sandy areas. If this threshold is exceeded, water and soil resources will be unbalanced, and the ecological environment will deteriorate. Accurate assessment of the suitable scale of cultivated land in semi-arid sandy areas is of great significance for sustainable utilization of cultivated land resources and regional ecological security. Most existing research methods are based on water resource constraints and rarely consider terrain factors. Therefore, based on the principle of water balance and with the Horqin Left Wing Rear Banner as the research area, this study adopted a multi-objective fuzzy optimization model and relative terrain index analysis method to explore the appropriate spatial ratio of irrigation and rainfed agriculture. The results show that the area of irrigated agriculture in the study area is 77,700 hm$^2$, and the appropriate scale is 91,700 hm$^2$. The current area of dry farming is 184,600 hm$^2$, and the suitable scale is 117,100 hm$^2$. The results also show that the utilization efficiency of water and soil resources in irrigated agriculture was not optimal, rainfed agriculture exceeded its suitable scale, and water and soil resources were seriously unbalanced. However, the region of cultivated land that exceeds the appropriate scale is mostly located in an area with poor terrain, less precipitation, and other unsuitable conditions for cultivation, which is prone to abandonment, resulting in deterioration of the ecological environment. Therefore, the spatial layout of agricultural land use in the study area should be adapted to local conditions, and the water-saving structure of irrigated agriculture should be optimized to achieve the maximum comprehensive benefits. Dry farming should be controlled on a reasonable scale, and the part exceeding the appropriate scale should be returned to grassland to ensure sustainable development.

**Keywords:** water resources; semi-arid sandy area; irrigated agriculture; rainfed agriculture; micro-topographical; multi-objective fuzzy optimization model

## 1. Introduction

The area of irrigated and rainfed agriculture in semi-arid sandy regions is restricted by water resources and terrain conditions [1–3], and a certain scale threshold exists [4]. The dune–interdune alternate geomorphology prevailing at a small scale in semi-arid sandy areas has a redistribution effect on regional water resources [5,6], leading to extensive variations in water conditions in different parts of the terrain, as well as different scales of irrigated and rainfed farming [7]. In recent years, with the rapid development of agricultural engineering technology, the extent of irrigated agriculture in semi-arid sandy areas has been expanded to pursue economic benefits [8–10], and irrigated water consumption continues to increase, easily leading to an imbalance in water and soil resources [11,12]. Rainfed farming is characterized by extensive planting, low yields, and abandonment in drought years. Extensive management and development easily fall into a vicious cycle of exploitation–abandonment–redevelopment–abandonment [13–15]. The suitability of the scale of irrigated and rainfed agriculture directly affects the stability of semi-arid

sandy regions [16]. Therefore, determining the appropriate spatial ratio of irrigated and rainfed agriculture in semi-arid sandy areas based on water resource constraints and micro-topography is particularly important.

Methods for calculating the suitable scale of cultivated land can be divided into five categories. The first category involves establishing a mathematical model [17–19]—that is, a functional relationship between water resources and the cultivated land area—by using the principles of water balance and of water and heat balance and subsequently calculating the suitable scale of regional cultivated land. For example, Ke Yingming et al., based on the principle of the water balance [20], and Li Jingxin et al., based on the principle of the water and heat balance [21], established models to calculate the regional, suitable arable land area. These methods emphasize the quantity of cultivated land but ignore optimization of the spatial distribution of cultivated land. The second category is based on multi-objective programming, which mainly leverages multiple objectives and constraints to establish equations and optimize the solution [22,23]. For example, for Shandong Province, Zhang Zhibin et al. determined constraint conditions according to planting benefits and water-saving benefits, established a multi-objective optimization model, adjusted the planting area of different kinds of crops, and maximized the comprehensive benefits of planting and water savings [24]. The third category is based on land suitability evaluations and GIS spatial analysis and relies on land suitability evaluations, the analysis of each constraint factor, and determination of the scale and spatial layout of agricultural land [25–28]. With these methods, when a water resource is taken as the constraint condition, indicators such as the river network density and distance from a water source are usually selected, and the availability of the water resource in a large area is used to constrain the agricultural land scale in a small local area, without considering coordination across the larger area. As a result, the calculated scale value is often too large. The fourth category uses the Soil and Water Assessment Tool (SWAT) model, the Cellular Automata (CA) model, and the Conversion of Land Use and its Effects at Small regional extent (CLUE-S) model for spatial optimization and determines the suitable scale for cultivated land [29–31]. These methods require more detailed data and parameter adjustments according to the actual conditions in the region. The fifth category uses the water footprint model [32–34]. A model for measuring the scale of cultivated land was established from the perspectives of the crop water footprint and the water source type. By the early 20th century, scholars had conducted many theoretical and practical studies on the water footprint model, providing a solid theoretical and practical foundation for determining the suitable scale of cultivated land. However, in China, studies of the water footprint began relatively late, and comprehensive quantitative research on the water footprint of the planting industry is lacking. Therefore, the suitable scale of cultivated land is mainly determined based on the water balance, multi-objective linear programming, suitability evaluations, and software models.

The above review shows that existing research is based on the size of water for irrigation agriculture, and the appropriate scale of the quantitative method for rainfed agriculture is not mature; for example, index selection is considered to be less suitable for dry land farming scale topographic factors, given factors such as rainfall. In fact, the blind large-scale reclamation of rainfed agriculture is the main cause of land degradation in semi-arid sandy areas; so, reasonably determining the appropriate scale of rainfed agriculture is very important.

Horqin Left wing Rear Banner is located in the eastern part of the agricultural–pastoral transitional zone in the core of the Horqin sandy land. This area was once a grassland with dense vegetation. The extensive management and development mode causes the agricultural development in this area to fall into a vicious cycle of development–abandonment–redevelopment–abandonment. Therefore, we selected Horqin Left Rear Banner as the research area. Based on the total amount of available agricultural water resources in the region, we studied the scale threshold and the optimization of the spatial layout of irrigated and rainfed agriculture to achieve the dual objectives of conserving resources

and promoting the sustainable and healthy development of agriculture in the semi-arid sandy area.

## 2. Materials and Methods

### 2.1. Study Area

Horqin Left Wing Rear Banner is located in the middle of the Horqin Sandy Land area (Figure 1), which is characterized by semi-arid sandy conditions and water resource shortages. The plain in the eastern part of the study area is a key grain-producing area, where corn and rice are primarily planted, and most of the central and western areas are key pastoral areas. In addition to the east and west Liaohe confluence in the east, an alluvial plain accounts for less than 3% of the total area, and other areas are interleaved with sandy meadows. The main reason for choosing this study area is that it is a typical semi-arid sandy land area in terms of resource conditions, ecological environmental conditions, and economic structures, making it an ideal object for this study. The results are as follows: (1) Horqin Left Wing Rear Banner is a semi-arid and semi-humid ecologically fragile area with an average annual precipitation of 414 mm, which is mainly concentrated in summer from June to August. The spatial difference in precipitation is large, with more precipitation in the east, west, and south and less precipitation in the north. (2) The typical topography of the study area is dune–interdune. (3) The current population of this region is 401,100, including 320,300 rural people. Residents live in poverty, and their livelihood largely depends on agricultural outputs and the ecological environment.

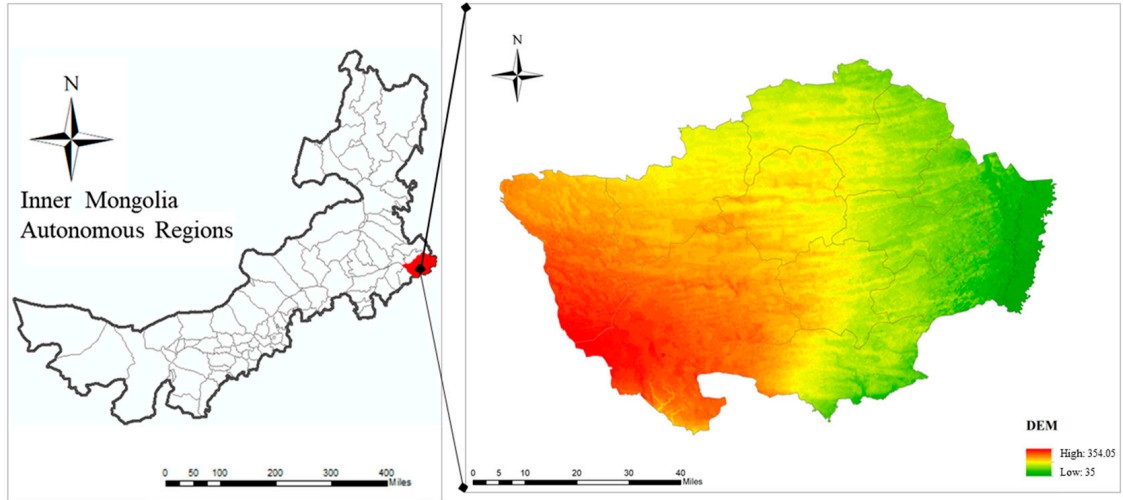

**Figure 1.** Overview of the study area.

### 2.2. Data Sources

The data included in this study are shown in Table 1.

**Table 1.** Data types and sources.

| Data Types | Data Sources |
| --- | --- |
| Elevation and other topographic data | ASTER GDEM 30 × 30 DEM data obtained from the China geographic data cloud website |
| Hydrological data | Tongliao Water Bureau, Horqin Left Wing Rear Banner sustainable development and utilization of water resources planning |
| Rainfall data | China Meteorological Data website (http://data.cma.cn/, (accessed on 15 January 2022)) |
| Land use data | Land consolidation planning of Tongliao City |
| Socioeconomic data | Tongliao statistical yearbook and field farmer survey |

## 2.3. Research Framework

The research presented in this paper is divided into the following three parts: (1) According to the available water resources in Horqin Left Wing Rear Banner, the effectiveness of the development of agricultural irrigated farming and economic, social, and ecological goals based on protecting the ecological environment, improving the efficiency of agricultural production, and increasing the economic efficiency were modeled according to the fuzziness in the planting industry system by using a multi-objective fuzzy optimization model. The model was established to calculate the suitable planting area of different types of crops; then, the suitable scale of irrigated agriculture was calculated. (2) Through a multi-objective land suitability evaluation and based on the micro-terrain conditions of the semi-arid sandy area, the concept of relative topography was introduced to calculate the suitable agricultural area for dry farming. (3) The suitable scale of irrigated and rainfed agriculture was calculated, the actual conditions of the semi-arid sandy area were discussed and analyzed, and reasonable suggestions for different landforms were proposed. The research framework is shown in Figure 2.

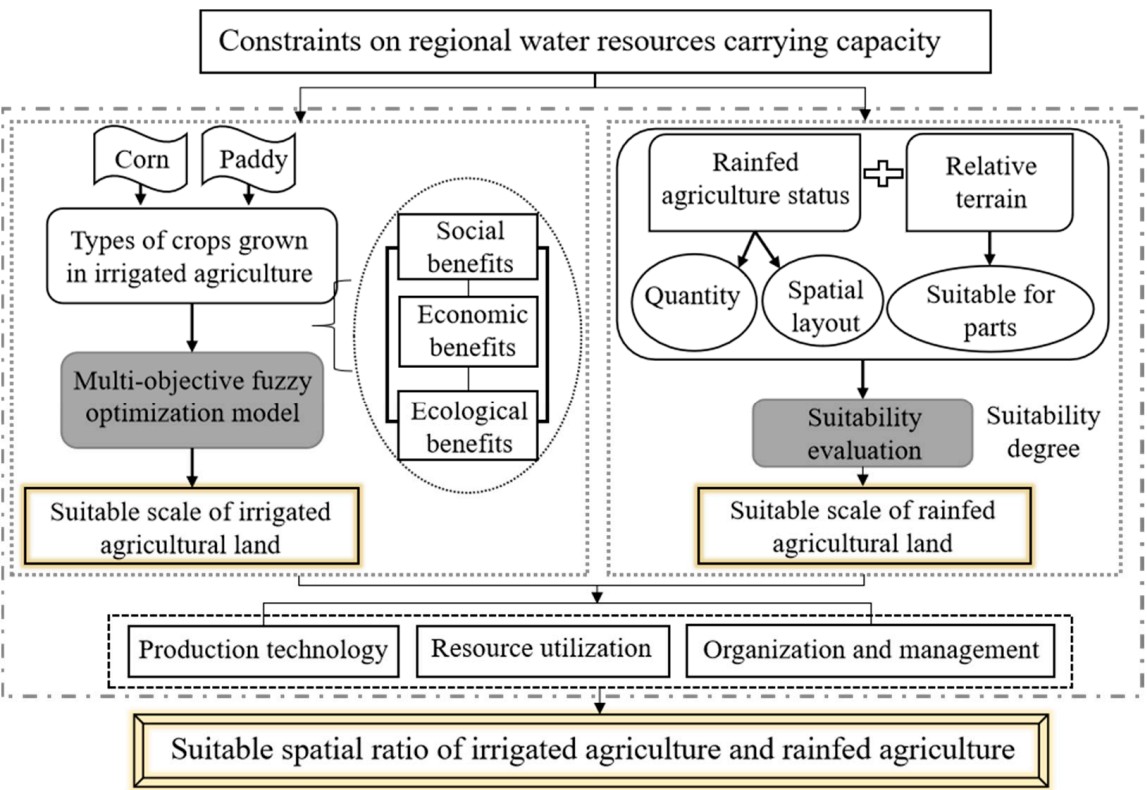

**Figure 2.** The research framework for the distribution of irrigated and rainfed agricultural land in the semi-arid sandy region.

## 2.4. Research Methods

### 2.4.1. Establishment of the Multi-Objective Fuzzy Optimization Model

The main crops grown with irrigation in the study area are rice and maize, and the total amount of available water resources in the region is limited. To obtain the maximum comprehensive benefits, the planting area of rice and maize should be reasonably controlled. The planting structure of rice and corn is affected by social, economic, ecological, and other factors, and no clear boundary exists for determining the planting area ratio, which is a fuzzy concept [35]. Therefore, a multi-objective fuzzy optimization model, the specific process of which is shown in Table 2, was adopted to determine the appropriate planting scale of rice and corn in the study area.

**Table 2.** Steps of building multi-objective fuzzy optimization model.

| | Metric | Formula | | Description |
|---|---|---|---|---|
| Step 1 | Establish a set | $x = \{x_1, x_2, \cdots, x_m\}$ | (1) | There are m kinds of crops in the irrigated area, which constitute the set $x$. |
| Step 2 | Determine the characteristic values of indicators | $x_j = \left(x_{1j}, x_{2j}, \cdots, x_{nj}\right)^T$ | (2) | The characteristics of sample j are represented by n index eigenvalues. |
| Step 3 | Index eigenvalue matrix | $x = \left(x_{ij}\right)$ | (3) | The sample set can be represented by an index eigenvalue matrix of order n×m, $x_{ij}$ is the eigenvalue of index i of sample j; $i = 1, 2 \ldots, n$. $j = 1, 2 \ldots, m$. |
| Step 4 | Relative membership transformation | $r_{ij} = \frac{x_{ij}}{\max x_j}$ <br> $r_{ij} = \min x_j / x_{ij}$ | (4) <br> (5) | The decision indicators of the crop planting structure are usually divided into two types, in which the larger the eigenvalue is, the better it is, or the smaller the eigenvalue is, the better it is. Equations (4) and (5) can be used to carry out relative membership transformation. |
| Step 5 | Relative membership matrix | $R = \left(r_{ij}\right)$ | (6) | The matrix of index eigenvalue (3) is transformed into a relative membership matrix. |
| Step 6 | Index relative membership degree | $r_j = \left(r_{1j}, r_{2j}, \cdots, r_{nj}\right)^T$ | (7) | The relative membership degree of m indexes of sample j is known from matrix R. |
| Step 7 | Index weight vector | $w_i = \dfrac{1}{1+\left\{\sum_{j=1}^{m}\left(1-w_{ij}\right)^p / \sum_{j=1}^{m}\left(w_{ij}\right)^p\right\}^{\frac{2}{p}}}$ | (8) | The $n$ indexes of the sample set have different influences on optimization; so, they should have different weights. Let the index weight vector $w = (w_1, w_2, \cdots, w_n)$ satisfy $\sum_{i=1}^{n} w_i = 1$. |
| Step 8 | Index weight matrix | $W = R^T = \left(W_{ij}\right)_{m \times n}$ | (9) | According to the concept that the membership degree can be defined as the weight in a fuzzy set, the relative membership degree matrix of the target to the fuzzy concept "importance" can be obtained by transposing matrix $R$. |
| Step 9 | Comprehensive benefits relative superiority degree | $u_j = \dfrac{1}{1+\left\{\sum_{i=1}^{n}\left[w_i\left(r_{ij}-1\right)\right]^p / \sum_{i=1}^{n}\left(w_i r_{ij}\right)^p\right\}^{\frac{2}{p}}}$ | (10) | Applying the two-level fuzzy optimization model, the optimal size of an optimum comprehensive benefit is obtained for crop $j$, where $P = 2$. |
| Step 10 | Objective function | $\max Z = \sum_{j=1}^{m} u_i s_i$ | (11) | $s_j$ is the optimal planting area of crop j, in hm$^2$. |
| Step 11 | Water constraint | $\sum_{j=1}^{n} g_j s_j \leq T$ | (12) | $g_j$ is the irrigated quota for crop i, in m$^3$/hm$^2$; $T$ is the total amount of available agricultural water resources in the study area. |
| Step 12 | Area constraint | $\sum_{j=1}^{n} s_j \leq S,$ <br> $\min s_j \leq s_j \leq \max s_j$ | (13) | $S$ is the total area of cultivated and irrigated land; $\min s_j$ and $\max s_j$ are the minimum and maximum planting areas, respectively, of crop j. |
| Step 13 | Non-negative constraint | $s_j \geq 0$ | (14) | The crop planting area in the study area cannot be negative. |

### 2.4.2. Relative Topography

Based on the unique geomorphic types in the study area, the relative topography formula was adopted to quantitatively describe the topographic relief characteristics in the local range of the region. According to Formula (15), to calculate the variation in the local terrain, the study area was divided into ten grades. According to the different levels relative to the terrain, the local topography in the study area was divided into four parts. The formula is as follows:

$$G_r = G_h - G_{mean} \tag{15}$$

where $G_r$ is the relative topography, $G_h$ is the elevation of a certain grid, and $G_{mean}$ is the average elevation of all grids within a certain neighborhood, with grid G as the center.

$G_{mean}$ was calculated with the focus statistics function of the neighborhood analysis tool in ARCGIS software; then, the terrain fluctuations within the local range were calculated using the Raster Calculator according to Formula (15).

### 3. Results

#### *3.1. Analysis of Irrigated Agricultural Land*

#### 3.1.1. Determination of the Superior Crop

All paddy fields in Horqin Left Wing Rear Banner were assumed to be planted with rice, and all irrigated fields were assumed to be planted with corn. The comprehensive benefits were evaluated based on the economic, social, and ecological benefits. The economic and social benefits are quantitative indexes, and their characteristic values are shown in Table 3.

**Table 3.** Characteristic values of the quantitative indexes.

| Crop Species | Economic Benefits (USD/hm$^2$) | Social Benefits (%) |
| --- | --- | --- |
| Rice | 1373.28 | 85 |
| Corn | 766.46 | 40 |
| Remarks | The economic benefit index value is the apportionment value of irrigated benefits. | The social benefit index value is the proportion of commodity outputs of agricultural products. |

The ecological benefit is a qualitative index, and its binary outcomes were evaluated for rice and maize in combination with relevant research results and practical investigations of semi-arid sandy areas. Considering these factors and water consumption, corn was found to be "more" important than rice.

According to Table 4,

$$r_{3j} = (0.538, 1.000) \tag{16}$$

**Table 4.** Characteristic values of qualitative indicators.

| Tone Operator | Same | Slightly | More | Obvious | Significant | Very | Extremely |
| --- | --- | --- | --- | --- | --- | --- | --- |
| Quantitative scale | 0.50 | 0.60 | 0.65 | 0.70 | 0.75 | 0.80 | 0.95 |
| Relative membership | 1.000 | 0.667 | 0.538 | 0.429 | 0.333 | 0.250 | 0.053 |

According to Formula (4) and Table 3,

$$R = \begin{bmatrix} 1.000 & 0.558 \\ 1.000 & 0.471 \\ 0.538 & 1.000 \end{bmatrix} \tag{17}$$

Then, the index weight matrix $W$ is obtained:

$$W = \begin{bmatrix} 1.000 & 1.000 & 0.538 \\ 0.558 & 0.471 & 1.000 \end{bmatrix} = (W_{ij})_{2 \times 3} \tag{18}$$

When P =2, according to Formula (8), the non-normalized weight vector $w_i =$ $(0.870, 0.814, 0.858)$, and the normalized weight vector of the index $w_i = (0.342, 0.320, 0.337)$. According to Formula (9), $u_j = (0.912, 0.769)$.

### 3.1.2. Determination of the Threshold of Constraint Conditions

At present, the total area of cultivated land in Horqin Left Wing Rear Banner is 262,000 hm$^2$, including 14,400 hm$^2$ of paddy fields and 63,000 hm$^2$ of irrigated land. According to the statistics released by the Tongliao Water Bureau, the total amount of available water resources for irrigated agriculture in Horqin Left Wing Rear Banner is 340 million m$^3$, all of which is obtained by exploiting groundwater. According to the industrial water quota standard of the Inner Mongolia Autonomous Region, when the irrigated guarantee rate is 75%, the irrigated quota for rice is 8300 m$^3$/ hm$^2$, and that for corn is 3200 m$^3$/ hm$^2$. Based on research on the demand for grain in China by Lu Liangshu et al., the minimum per capita grain demand in Horqin Left Wing Rear Banner is 400 kg/a, and the proportion of rice is 55.4%; thus, the minimum per capita rice demand in the study area was estimated to be 221 kg/a. According to the statistical yearbook, the current population size of Horqin Left Wing Rear Banner is 401,100. The yield of rice is 9.77 t/ hm$^2$, and the minimum rice demand based on food security is 0.9100 hm$^2$. Due to substantial evaporation in semi-arid sandy areas, paddy planting consumes a large amount of water resources, restricting the paddy planting area under current situations. The area is not expected to increase.

### 3.1.3. Determination of the Multi-Objective Fuzzy Optimization Model

According to Formula (10), the multi-objective fuzzy optimization model of irrigated agricultural land in Horqin Left Wing Rear Banner is

$$\max Z = 0.912s_1 + 0.769s_2 \tag{19}$$

The constraints are as follows:

$$\text{Water constraint}: \ 8300s_1 + 3200s_2 \leq 3.40 \times 10^8 \tag{20}$$

$$\text{Area constraint}: \ s_1 + s_2 \leq 26.20 \times 10^4, \ 0.91 \times 10^4 \leq s_1 \leq 1.44 \times 10^4 \tag{21}$$

$$\text{Non-negative constraint}: \ s_1 \geq 0, \ s_2 \geq 0 \tag{22}$$

Using linear programming to solve the model, the optimal irrigated agricultural land area in Horqin Left Wing Rear Banner was determined to be 91,700 hm$^2$, including 10,100 hm$^2$ for rice and 82,600 hm$^2$ for corn.

### 3.2. *Analysis of Rainfed Agricultural Land*

With an increase in grade, the relative topography of the area increased, as shown in Figure 3.

According to the actual conditions in the study area and the present distribution of cultivated land, prioritizing irrigated agriculture is suitable for rainfed agricultural land located in the transition zone from irrigated agricultural land to upper sand mounds or low mounds, specifically, from relative area topography grades of 5 to 7. The region covers an area of 117,100 hm$^2$; so, the proper scale of rainfed agriculture is 117,100 hm$^2$.

### 3.3. *Optimization of the Spatial Distribution of Irrigated and Rainfed Agricultural Land*

Agricultural development in the study area is strongly restricted by water resources and presents some differentiated spatial distribution characteristics in combination with the micro-topography. Therefore, agricultural land use should match local conditions in terms of the spatial layout. The typical dune–interdune topography in the study area can be divided into four parts, as shown in Figure 4.

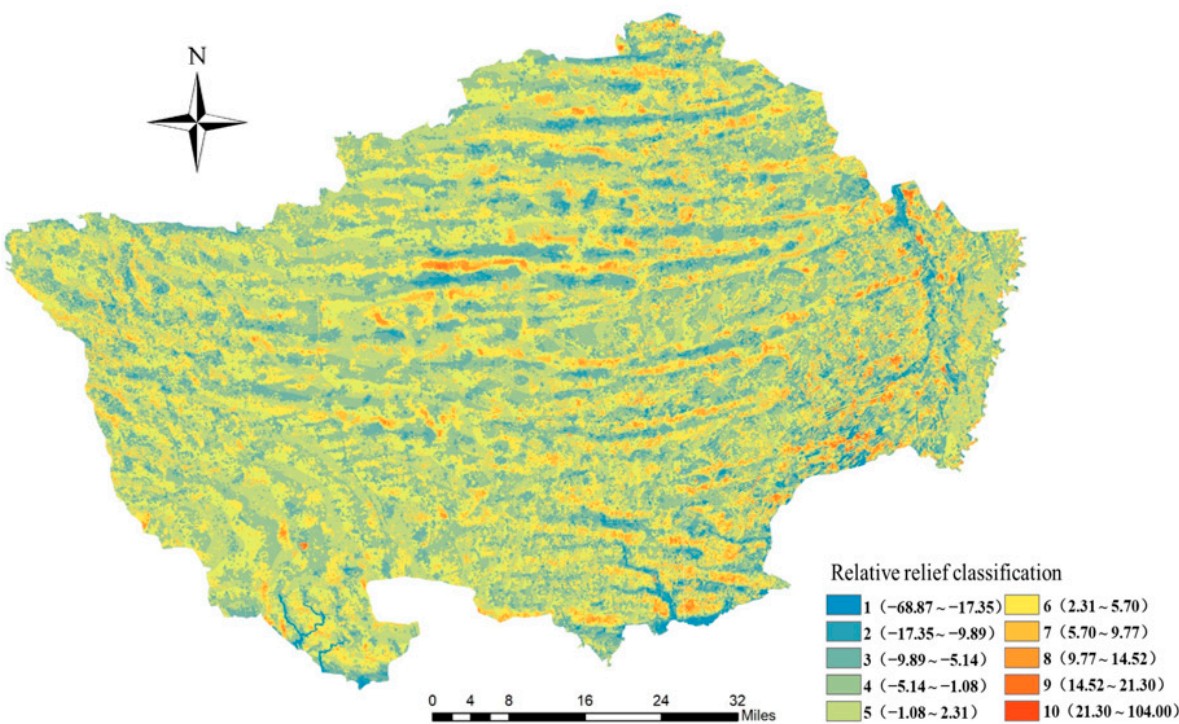

**Figure 3.** Relief classification map of the study area based on relative topography.

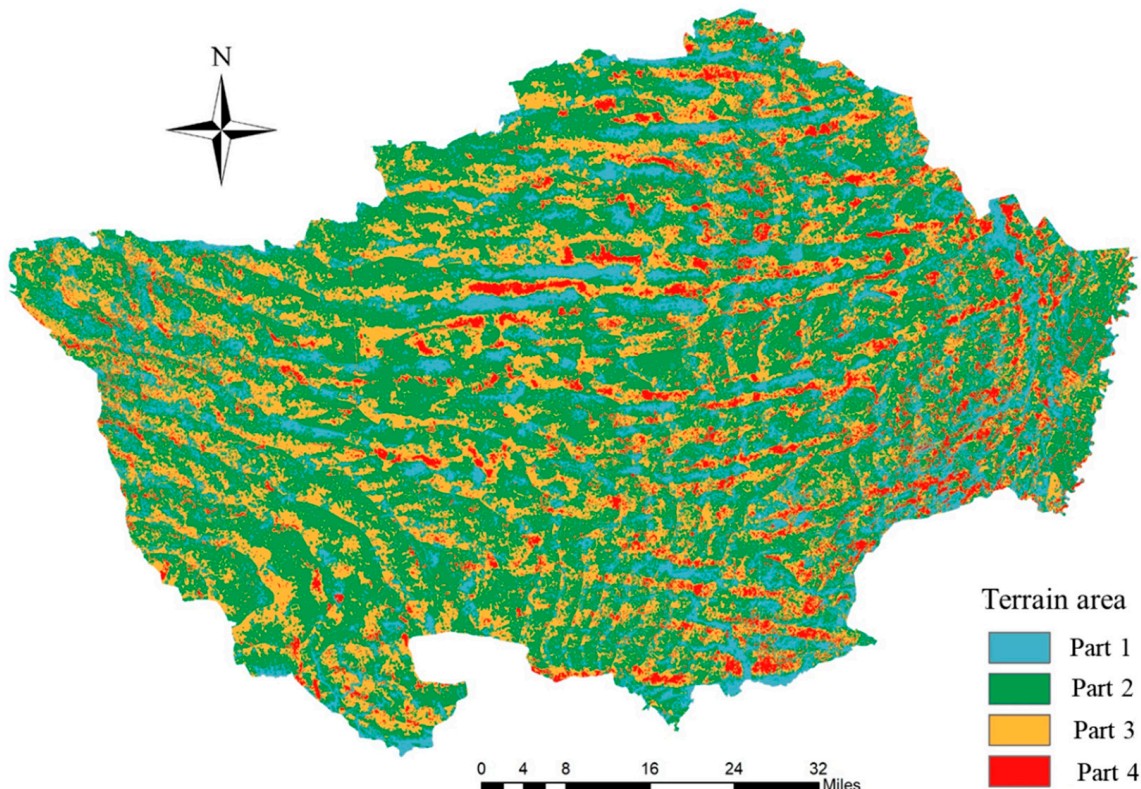

**Figure 4.** Topographic distribution map of the study area.

Due to the alternating dune–interdune micro-geomorphology, different measures should be used in different terrain areas (Figure 5). Part 1 is low-lying and has a large amount of water, and crops are greatly affected by flood disasters. Due to annual variation in precipitation, crops cannot generate a stable income; so, the area is not suitable

for agriculture. It should be mainly protected to reduce disturbances of the ecological environment caused by development, and this part should be designated as ecological land. Part 2 has a gentle terrain, relatively high groundwater level, and the best natural conditions. It is usually distributed in contiguous regions and covers a large area, making it suitable for the development of large-scale irrigated agriculture. Part 3 is in the transition zone from irrigated agricultural land to upper sand mounds or low mounds with slightly worse natural conditions and a higher terrain, making it unsuitable for the development of irrigated agriculture. However, small-scale land leveling can be carried out to develop rainfed agriculture. Part 4 is the middle and upper parts of the transition zone, with the worst moisture conditions and serious desertification, making it unsuitable for the development of agriculture. It can be classified as ecological land for improvement and protection to avoid the negative effects of desertification on local villages and farmland.

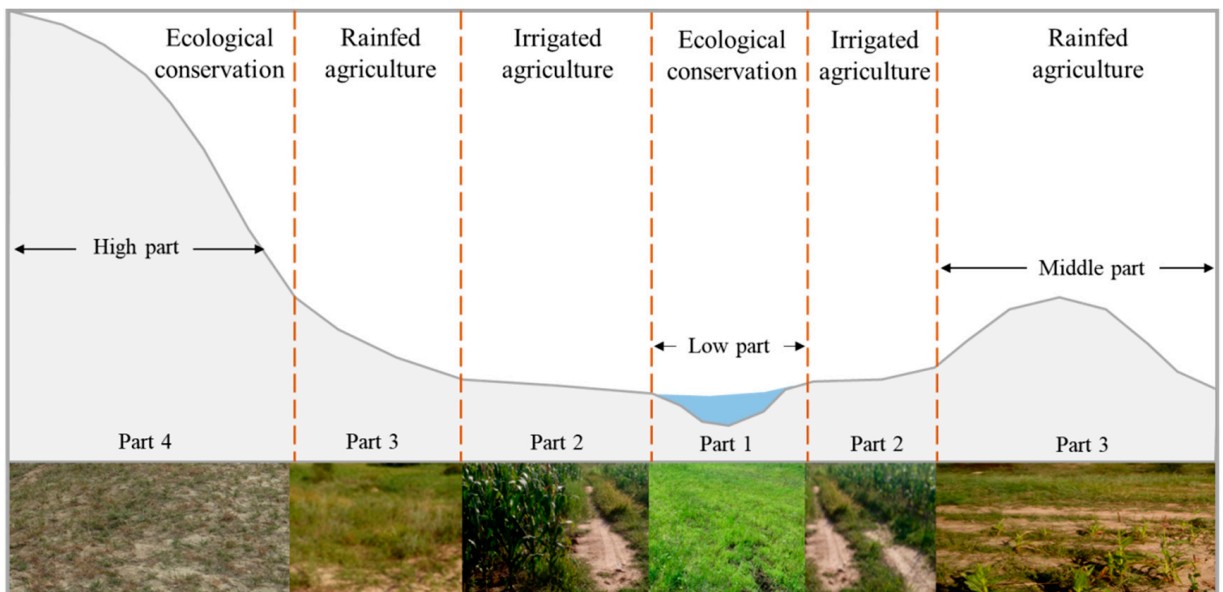

**Figure 5.** Spatial distribution patterns of irrigated and rainfed agricultural land in the semi-arid sandy region.

Due to population growth, the demand for cultivated land is increasing, irrigated and rainfed agricultural areas continue to expand, and cultivated land is gradually expanding from Part 2 to Part 1, Part 3, and even Part 4. From the perspective of regional ecological security, Part 1 and Part 4 should not be used for cultivation or be exposed to excessive human disturbance, and ecological protection should be the focus. Parts 2 and 3 can be developed as agricultural land but on a suitable scale based on the water resource carrying capacity.

## 4. Discussion

Land use in semi-arid sandy areas is restricted by water resources. If the intensity of land use is too small, the needs of local residents for survival and development cannot be met. If the intensity of utilization is too large, the water resource carrying capacity will be exceeded and the local ecological environment will be damaged. Therefore, reasonably determining the suitable development and utilization intensity and accurately calculating the suitable scale of agricultural land in the semi-arid sandy area are of great significance to realize the sustainable utilization of regional land resources. Most existing research has focused on areas with concentrated contiguous cultivated land, such as river basins and plains, while few studies have focused on semi-arid sand areas with fragile ecological environments and undulating terrain [36,37]. Most research methods are based on the availability of water resources to determine the scale of irrigation and agricultural

land [38–41]. For semi-arid sand areas with water shortages, irrigated agriculture and rainfed agriculture are common. Irrigated agriculture is largely dependent on the total amount of water available for irrigation, while rainfed agriculture is mainly dependent on natural precipitation. The landform types among dunes in semi-arid sand areas have a role in the redistribution of regional water resources, resulting in great differences in water conditions in different terrains. In semi-arid sandy areas, the topography is an important factor that affects the scale and layout of rainfed agricultural land, which has not been fully considered in existing studies [42,43]. Therefore, we developed a calculation method that is suitable for the scale of agricultural land in semi-arid sandy areas. According to different utilization modes, agricultural land was divided into irrigated agriculture and rainfed agriculture, and the appropriate scale of each was calculated. Aiming at the fuzzy characteristics of irrigated agriculture systems, a multi-objective fuzzy optimization model was established to calculate the suitable planting area of different types of crops; then, the suitable scale of irrigated agriculture was calculated. According to the micro-terrain conditions of semi-arid sandy areas, the concept of relative topography was introduced to calculate the suitable scale of rainfed agriculture.

As a typical representative of semi-arid sandy areas, Horqin Left Wing Rear Banner has been subjected to wind erosion and desertification for a long time, and the dune landform types are generally undulating. Such local terrain conditions redistribute water and heat resources in the region, resulting in large differences in the water conditions and land use suitability in different areas. In this study, the local terrain was divided into four parts, where low flat areas with better natural conditions were found to be suitable for the distribution of agricultural land. The field investigation showed that the extensive utilization mode of wide planting and low yields is widespread in the Horqin Left Rear Banner area, and a large amount of grassland has been reclaimed for arable land. The results show that the actual dry farming area in the study area far exceeds the suitable scale of dry farming. However, the area of cultivated land that exceeds the appropriate scale is mostly located in an area with poor terrain, less precipitation, and other unsuitable conditions for cultivation; most of this area is unstable cultivated land, which is prone to abandonment, resulting in deterioration of the ecological environment and aggravating the contradiction between production and ecology. Therefore, the development and utilization intensity of cultivated land should be reasonably controlled according to the suitable scale of irrigation and dry farming in Horqin Left Wing Rear Banner. For terrain that is not suitable for cultivation, ecological protection should be given priority, and human interference activities should be reduced. At the same time, the scale of agricultural land should be controlled within a reasonable range based on the water resource carrying capacity.

In this study on sustainable agricultural development in semi-arid sandy areas, full consideration was given to water and terrain conditions, the appropriate scales of irrigated and rainfed agriculture were determined, and the physiognomy types of land use patterns were inferred according to the geological characteristics of the area and the inherent qualities of the agricultural land. The contradiction between water and soil resources is prominent in semi-arid sandy areas. Agricultural development is constrained by the water resource threshold and presents certain differentiated distribution characteristics in combination with micro-topographical conditions. Therefore, the spatial layout of agricultural land use should adhere to local conditions. In different landforms, due to the different amounts of water resources, the scale and type of agriculture that can be carried are also different. In low-lying places, water is more abundant, but crops are greatly affected by floods and waterlogging disasters. Due to different annual precipitation levels, obtaining a stable income from crops is difficult. The higher the terrain is, the poorer the water conditions are, the more fragile the ecological environment is, and the more easily the land is degenerated. In the face of the present situation of agricultural development in semi-arid sandy areas, realizing the gradual withdrawal of low-grade suitable arable land and the sustainable, orderly, and efficient development and utilization of high-quality land is key. Future research will focus on land development and the utilization intensity in semi-arid

sandy areas by integrating ecological security networks from the perspective of ecological security to define agricultural land use patterns that are adapted to the characteristics of regional human–land systems and high-quality development to promote the harmonious development of such systems in semi-arid sandy areas.

## 5. Conclusions

With Horqin Left Wing Rear Banner as the study area, we explored the scale threshold and the optimization of the spatial layout of irrigated and rainfed agricultural land by using relative topography analysis to provide a scientific basis and decision-making reference for promoting the sustainable and healthy development of ecology and agriculture in semi-arid sandy areas. The main conclusions are as follows:

(1) The irrigated agricultural land area is 77,700 hm$^2$ in Horqin Left Wing Rear Banner, including 14,400 hm$^2$ of paddy fields and 63,000 hm$^2$ of irrigated land, and the water resource carrying capacity has not yet been exceeded. However, the current planting structure has not yielded the maximum comprehensive agricultural benefit and should be adjusted to appropriately reduce the area of rice, which has a high rate of water consumption, and increase the area of corn, which has a low rate of water consumption. The area of suitable irrigated agricultural land in this region was found to be 91,700 hm$^2$, that of rice paddy fields was found to be 10,100 hm$^2$, and that of irrigated farmland was found to be 82,600 hm$^2$.

(2) The rainfed agriculture in Horqin Left Wing Rear Banner is mainly characterized by extensive planting, low yields, and extensive operations. In drought years, the land is abandoned and falls into a vicious cycle of development–abandonment–redevelopment–abandonment, which is the main reason for land degradation in this area. The suitable scale of rainfed agricultural land in the study area is 117,100 hm$^2$, and the actual area of such land is 184,600 hm$^2$. The rainfed agricultural land exceeding the suitable scale is mostly unstable arable land, which is prone to abandonment and should be returned to grassland.

Based on terrain analysis, the study area was divided into four parts. From the perspective of regional ecological security, irrigated agriculture is suitable for distribution in part II, dry farming is suitable for distribution in part III, and parts I and IV are unfavorable for cultivated land use and human activity and should be given priority for ecological protection. However, due to the increasing demand for cultivated land, cultivated land gradually expands from part II to part I, part III, and even part IV. However, the increased cultivated land area is mostly located in areas unsuitable for farming, such as those with poor topography and less precipitation, which are easy to be abandoned, leading to deterioration of the ecological environment. Therefore, the spatial layout of agricultural land in semi-arid sandy areas should be adapted to local conditions. Under the constraints of water resources and terrain conditions, the appropriate scale of irrigation and dryland agriculture in semi-arid sandy areas should be clarified, and the land use pattern oriented to the special landform types in semi-arid sandy areas should be constructed.

**Author Contributions:** Y.X. designed the research; H.Z. performed the data analysis and wrote the main manuscript text; Z.S. and K.W. wrote the main manuscript and created the figures and tables; Z.C. provided conceptualization, investigation, and methodology. All authors have read and agreed to the published version of the manuscript.

**Funding:** This research was supported by the Third Xinjiang Scientific Expedition Project of the Ministry of Science and Technology "Investigation on the Status quo and Evolution of Land Resource Utilization in the Tarim River Basin" [grant number 2021xjkk0202] and the Special Scientific Research of the Ministry of Land and Resources of China—Key Technology and Demonstration based on Protective Development of Sandy Land in Inner Mongolia [grant number 201411009].

**Institutional Review Board Statement:** The authors declare that the submitted manuscript is original and unpublished elsewhere and that this manuscript complies with the Ethical Rules applicable for this journal.

**Informed Consent Statement:** The authors give their permission to participate. The authors consent to publish this article in your journal and to transfer copyright to the publisher once the paper has been accepted.

**Data Availability Statement:** The datasets generated during the current study are available from the corresponding author on reasonable request.

**Acknowledgments:** We are grateful for the comments and criticisms of the journal's anonymous reviewers and our colleagues.

**Conflicts of Interest:** The authors declare no conflict of interest. The opinions expressed here are those of the authors and do not necessarily reflect the position of the government of China or of any other organization.

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
