# Peer review of "Distribution of Irrigated and Rainfed Agricultural Land in a Semi-Arid Sandy Area"

_land, doi:10.3390/land11101621_

Round 1
Reviewer 1 Report
The authors propose a study that evaluates the scale threshold and spatial distribution of irrigated area and rainfed agriculture in a semi-arid sandy area of Northern China. The study is potentially interesting, but I have several concerns about the paper.
1. The authors did not properly justify the novelty compared to the published literature
2. The abstract has an uncommon structure
3. The representativeness of the case study is not shown in details
4. Some methodological sentences are reported in the Results section
5. Some tables and figures should be improved
6. The Discussion section is poor, since it lacks cross-comparisons with relevant studies and no precise links with the Results section are given.
I am not a native English speaker, but I feel that the language could be improved.

Author Response
Dear reviewer:
Thanks very much for your time to review this manuscript. I really appreciate all your comments and suggestions. We have considered these comments carefully and tried our best to address every one of them.

Reviewer 2 Report
Dear Authors,
I really appreciated in your article the analysis of the literature, especially in relation to the different methods of analysis that you used and I found the article very balanced, that is, the right length, essential things etc.. I think that in general the article is well done but it needs some changes in different sections, therefore, I provided some general and specific suggestions that I hope will help you.
Regarding the abstract, please revise it avoiding to point out the results and the rest, it's now too schematic and you don't need to put all that data in the abstract, so please arrange it in a more fluent way and check also the interline spaces.
The bibliographic citations are done in the classic way (Names of authors and year) and not as required by MDPI (number in square brackets which is then reported in the bibliography) so please modify them.
Regarding the study area, I think you must provide some more data, such as extension, population, importance of the agricultural sector. Furthermore, you should say something about if it is there a demand / pressure from dry farmers for water and prove it with some additional info.
· Line 4: I believe is wrong the way you put the references to the authors, the numbers should be superscripts
· Line 8: please delete “correspondence:”
· Line 50: in the sentence “The first is ...”, “the” goes is lowercase because it is after “:”
· Line 74 and in general, the meaning of the abbreviations should be fully explained the first time they are found
· Line 109: the title of the figure I would put it in a smaller font
· In general, please revise the numbering of tables and figures, for example in the text you put 3-1, 3-2 because they are the first and second tables of paragraph 3, but then in the title there is only “table 3” for both, as well as the figures, in the title is always written “figure 3” and I think the reader get confused in this way
· line 158 and following, please check the sentences, maybe I misunderstood but, for corn, firstly you talk about dry land and then about irrigated, is not clear for me, please revise it
· table 3: in addition to Yuan, you should also provide the dollar value, perhaps in a footnote
· line 165 and in general, leave a line/space between the table and the text and within the text
· it seems to me that formulas 6 and 7 have a larger character, please standardize
· line 209: put the “,” among the numbers, for example 117100 must be 117.100
· line 211 this paragraph should be 3.3 and not 3.2
· line 215: “A according” should be “According”
· line 251 I think it must be “in Horqin ...”
· line 255 after “evaluation” goes the period, not the comma, because with Furthrmore it starts another sentence
· In the paragraph of the discussions perhaps some bibliographical references should be added
· In the final part of the conclusions, you repeated things already said in paragraph 3 so please revise it
Author Response

(The authors gave the same response as above.)

Round 2
Reviewer 1 Report
The authors have fairly replied to the reviewers' comments and now the paper is improved.